# Effect of Visual Booklets to Improve Bowel Preparation in Colonoscopy: Systematic Review with Meta-Analysis

**DOI:** 10.3390/jcm12134377

**Published:** 2023-06-29

**Authors:** Giuseppe Losurdo, Maria Ludovica Martino, Margherita De Bellis, Francesca Celiberto, Salvatore Rizzi, Mariabeatrice Principi, Enzo Ierardi, Andrea Iannone, Alfredo Di Leo

**Affiliations:** 1Section of Gastroenterology, Department of Precision and Regenerative Medicine and Ionian Area, University of Bari, Piazza Giulio Cesare 11, 70124 Bari, Italy; ludovicamarti@gmail.com (M.L.M.); margideb@libero.it (M.D.B.); salvatore.rizzi.med@gmail.com (S.R.); b.principi@gmail.com (M.P.); ierardi.enzo@gmail.com (E.I.);; 2Course in Organs and Tissues Transplantation and Cellular Therapies, Department of Precision Medicine Jonic Area, University “Aldo Moro” of Bari, 70124 Bari, Italy

**Keywords:** colonoscopy, bowel preparation, booklets, visual, PEG, quality

## Abstract

An optimal bowel preparation for colonoscopy is essential to increasing the quality of the examination. Visual booklets have been proposed with conflicting results to enhance bowel preparation. A literature search was performed in March 2023 in the most important databases. Only RCTs were selected. We calculated odd ratios (OR) for dichotomous outcomes. Mean differences (MD) or standardized mean differences (SMD) were used for continuous outcomes. We estimated heterogeneity with the Chi^2^ and the I^2^ statistics. In cases of high heterogeneity, a random effect model was used. Six studies were selected, enrolling 1755 patients overall. Adequate bowel preparation was observed in 86.7% of the booklet group versus 77.5% of the control group, with an OR = 2.31 in favor of the booklet. In studies using a 4-L PEG-based preparation, no difference compared to controls was observed, while in non-PEG formulations, preparation with booklets was better than in controls (OR = 5.10, 95% CI 1.82–14.27, *p* = 0.002). Two studies were performed in an inpatient setting without any differences between booklets and controls, while outpatients receiving booklets had better results (OR = 7.13, 95% CI 5.39–9.45, *p* < 0.001). The adenoma detection rate was similar between the two groups. In conclusion, booklets are useful to improve bowel preparation. Outpatient settings and preparations not containing PEG could benefit more from booklets.

## 1. Introduction

Colonoscopy is the gold standard for diagnosing and treating colorectal diseases. It is widely available, has high sensitivity and specificity for investigating colonic diseases, and allows tissue sampling for achieving pathological diagnosis and, furthermore, therapeutic procedures [1]. The success of a colonoscopy is highly dependent on the quality of the bowel preparation. Adequate bowel preparation can minimize the risk of prolonged procedure time, incomplete procedures, repeated examinations, missed lesions, and delayed diagnosis, thus avoiding the waste of medical resources [2,3]. Consistent evidence revealed that adequate bowel preparation provides good colonoscopy vision and thus increases the detection rate of colonic polyps and adenomas [3,4]. Previous studies have determined several variables that can influence the quality of bowel preparation, such as appropriate dietary restrictions and proper administration of preparation solutions [5]. In addition, non-medical patient factors, including low education level, low socio-economic status, longer appointment waiting time, low health literacy (i.e., patient’s capability of acquiring and understanding general health knowledge and services), and low patient activation (motivation to employ themselves in health promotion) are likely to result in non-compliance with bowel preparation instructions [6]. Adequate comprehension of the preparation and colonoscopy details is a critical contributor to adequate bowel cleansing. Patients usually receive written or verbal instructions from professionals before colonoscopy for details regarding preparation and dietary restrictions, which are defined as standard patient instructions [7]. However, research by Ness et al. found that such guidance often fails to achieve satisfactory bowel preparation [8]. To increase the awareness of bowel preparation in outpatients and improve compliance, researchers have made extensive attempts such as visual aid and cartoon education booklets [7]. Educational booklets not only show details about the dietary and purgative instructions, but they also give a more comprehensive explanation of bowel preparation. Most educational booklets and visual aids, such as pictures and cartoons, are more concise and intuitive than written and oral instructions [6].

The aim of this meta-analysis is to evaluate the efficacy of visual aids, booklets, and pamphlets compared with standard patient instructions to obtain an adequate bowel preparation. We have considered multiple aspects influencing a good colonoscopy and we individually analyzed them (quality bowel preparation and preparation according with type and setting, adenoma detection rate, insertion, and withdrawal time).

## 2. Materials and Methods

The present review was performed according to the Preferred Reporting Items for Systematic Reviews and Meta-Analyses (PRISMA) indications [9].

### 2.1. Data Sources and Searches

A literature search for randomized controlled trials (RCTs) comparing leaflets to standard care in people undergoing colonoscopy was performed in March 2023 in PUBMED, SCOPUS, ScienceDirect, the Cochrane Central Register of Controlled Trials (CENTRAL), and ClinicalTrials.gov. The following string was used: “colonoscopy preparation OR bowel preparation OR cleansing AND (booklet OR brochure OR depliant OR visual aid OR leaflet)”. Articles in languages other than English and conference abstracts were excluded. Relevant articles were also searched from the bibliography of selected articles and guidelines.

### 2.2. Study Selection

Two reviewers (MLM and MDB) separately assessed the searches by title and abstract, then the full text, to look for potentially eligible trials. Uncertainties were solved by a third reviewer (GL). We included any randomized controlled trials that compared visual, leaflets, or booklets given in addition to standard of care compared to standard of care alone to explain bowel cleansing for colonoscopy. The booklet had to contain figures resuming the modality of preparation intake and pictures of optimal stools to achieve for adequate cleansing. We excluded other studies in which other informative methods, such as videos, phone apps, messages, web-based platforms, or phone calls were used.

### 2.3. Data Extraction and Quality Assessment

Two authors (MLM and MDB) independently evaluated any RCT for eligibility and extracted data on study, population, intervention (experimental and comparator treatments), and outcomes features. Uncertainties were solved by a third reviewer (GL).

### 2.4. Outcomes Measures

The primary outcome was adequate bowel preparation, evaluated with any validated scale for the assessment of bowel cleansing.

Pre-specified secondary outcomes were quality of bowel preparation measured as continuous parameter, quality of bowel preparation per colonic segment (i.e., right colon, transverse, left colon), adenoma detection rate (ADR), polyp detection rate, number of adenomas per patient, number of polyps per patient, cecal intubation rate, insertion and withdrawal time, any adverse events, and willingness to repeat the preparation and colonoscopy.

### 2.5. Risk of Bias Evaluation

We evaluated the study level risk of bias using the Cochrane risk of bias tool, which includes the domains of random sequence generation, allocation concealment, blinding of participants or investigators, blinding of outcome assessment, completeness of outcome data, selective reporting, and other limitations to validity.

### 2.6. Data Synthesis and Analysis

Odd ratios (OR) with 95% confidence intervals (95%CI) were estimated for dichotomous outcomes. Mean differences (MD) or standardized mean differences (SMD) with 95%CI were calculated for continuous outcomes, if unit of measurements were homogeneous or not across studies, respectively. We assessed pooled estimates using the DerSimonian and Laird random-effects model [10].

We formally estimated the heterogeneity of intervention effects among studies with the Chi^2^ (Cochrane Q) and the I^2^ statistics. In case of high heterogeneity, a random effect model was used, otherwise a fixed effects model was preferred. The fixed effects model was also applied if the analysis enclosed less than three studies [11].

In case of heterogeneity, subgroup analyses were performed. In detail, we planned pre-specified subgroup analyses by type of bowel preparation (Polyethylene glycol—PEG—vs. others) or clinical setting (inpatients vs. outpatients). We estimated differences among subgroups using the Mantel–Haenszel test.

We planned to explore publication bias with funnel plots, where at least 10 studies were included [12].

We rated the quality of evidence according to the Grades of Recommendation, Assessment, Development, and Evaluation (GRADE) approach [13].

All analyses were carried out using RevMan version 5.4.1 (The Cochrane Collaboration, Copenhagen, Denmark).

## 3. Results

### 3.1. Characteristics of Included Trials

The literature search retrieved 3815 studies. After duplicate elimination and exclusion of non-pertinent articles, we selected six eligible RCTs [14,15,16,17,18,19], enrolling 1755 patients (857 in the booklet and 898 in the standard care group) (Figure 1).

Table 1 shows the characteristics of the populations and interventions in the included studies. Three (50%) out of six studies were performed in the USA [14,15,18]. The Boston bowel preparation scale was used in four (67%) studies [14,15,16,19]. Adequate preparation was considered for Boston if total points ≥6 with at least ≥2 point for each bowel segments and, for Likert, if ≥3 points. All preparations were administered as a “same day” regimen, except for [15,19] (split in all patients) and Guardiola-Arevalo (split in the 30%) [15]. PEG was the most used agent [14,15,16,19], while other studies used phosphate or magnesium citrate.

### 3.2. Quality of Included Trials

Regarding the tool for assessing bias, reported in Appendix A, only one study scored several “high risk” items, regarding performance, detection and reporting bias. Detection bias was insufficiently reported in two articles.

### 3.3. Outcomes

Summary of outcome results and quality of evidence appraised with the GRADE approach are reported in Table 2.

#### 3.3.1. Quality of Bowel Preparation

Adequate bowel preparation was evaluated in six studies. It was obtained in 86.7% (743 out of 857) in the booklet preparation group versus 77.5% (696 out of 898) in the control group. There was a higher likelihood of achieving adequate bowel cleansing in the booklet group compared to the standard care group (OR 2.31, 95% CI 1.20 to 4.45, *p* = 0.01) (Figure 2). A random effects model was used due to presence of heterogeneity (Chi^2^ = 23.16, I^2^ = 78%).

The total preparation scores were assessed in five studies [14,15,16,18,19], enrolling 1627 patients. There was a higher mean score in the booklet group compared to the standard care group (SMD 0.21, 95% CI −0.04 to 0.47, *p* = 0.10), although this different did not achieve statistical significance (Figure 3). A random effects model was used due to the presence of heterogeneity (Chi^2^ = 19.07, *p* = 0.0008, I^2^ = 79%).

Sub-analysis of preparation score per each colonic segment was available only for two studies [15,16]. Booklets were more effective than the control in the transverse colon (MD = 0.37, 95% CI 0.01–0.72, *p* = 0.03), while no differences were found in the right colon (MD = 0.27, 95% CI −0.09–0.64, *p* = 0.14) and left colon (MD = 0.34, 95% CI −0.03–0.71, *p* = 0.07). MD was used because both studies used the Boston scale. Since only two studies were analyzed, a fixed effects model was used. Forest plots are depicted in Figure 4.

#### 3.3.2. Quality of Preparation According to the Type of Preparation

In four studies, a four liter PEG-based preparation was used [14,15,16,19]. The pooled analysis did not find any difference compared to controls (OR = 1.58, 95% CI 0.94–2.97, *p* = 0.15). Random effect was used because of high heterogeneity (Chi^2^ = 9.02, *p* = 0.03, I^2^ = 67%, Figure 5a). Different formulations were used by Ozkan (Sodium phosphate + enema) [17] and Spiegel (Magnesium citrate, sodium phosphate, PEG) [18], and, in this case, preparation with booklets was superior to controls (OR = 5.10, 95% CI 1.82–14.27, *p* = 0.002).

#### 3.3.3. Quality of Preparation According to Setting

Two studies, performed in an inpatient setting [15,16] did not show any difference between booklets and controls (OR = 1.22, 95% CI 0.69–2.17, *p* = 0.49). In outpatients, on the other hand, patients receiving booklets had better results than controls (OR = 7.13, 95% CI 5.39–9.45, *p* < 0.001). Fixed effect was used for this analysis. Forest plots are reported in Figure 5b.

#### 3.3.4. Adenoma Detection Rate

ADR results were reported in three studies [14,15,19]. ADR was similar between the two groups (Figure 6a), with an OR = 0.93 (95% CI 0.75–1.16, *p* = 0.53).

The mean number of polyps per patient (Figure 6b) did not differ between the two groups (MD = 0.22, 95% CI −0.14–0.58, *p* = 0.23).

The fixed effects model was applied in both analyses.

#### 3.3.5. Insertion and Withdrawal Time

Insertion and withdrawal time were expressed in minutes and two studies investigated this finding [14,19]. Regarding insertion time, no difference between the two groups was detected (MD = 0.54, 95% CI −0.59–1.67, *p* = 0.35, Figure 7a).

On the other hand, a longer withdrawal time was observed in controls, with a MD = −1.74 (95% CI −2.93 to −0.56, *p* = 0.004), as shown in Figure 7b. The fixed effects model was used for such analyses.

Regarding other outcomes, cecal intubation rate was described in only one study [16], showing no statistical difference between the two groups. Adverse events ratio was recorded only in one trial [14], with similar prevalence in both groups. Willingness to repeat preparation and colonoscopy was not reported in any study.

## 4. Discussion

In the present meta-analysis, we showed that adequate bowel preparation was achieved in 86.7% of the booklet group versus 77.5% of the control group, with an OR = 2.31 in favor of booklets. In studies on non-PEG formulations, preparation with booklets was better than in the controls (OR = 5.10). Furthermore, in an inpatient setting, we did not detect any difference between booklet and controls, while outpatients receiving booklets were better prepared (OR = 7.13. Adenoma detection rate was similar between the two groups. Colonoscopy plays an important role in the CRC screening program and was proved to decrease the incidence and mortality of CRC [20]. Adequate bowel preparation is pivotal for complete mucosal inspection in order to increase the detection of precancerous lesions [21]. Indeed, it has been proven that poor colon preparation decreases ADR significantly. Few studies have evaluated the miss rate of colonic polyps due to inadequate cleansing. Chokshi et al. found an adenoma miss rate of 48% in repeat colonoscopy in patients with prior inadequate preparation [22]. Lebwohl et al. revealed adenoma and advanced adenoma miss rates of 42% and 27%, respectively, on 1-year repeat colonoscopies [23]. Of note, both studies had limitations derived from the single-center retrospective design. Furthermore, a screening colonoscopy performed in the presence of inadequate bowel preparation is suggested to be repeated [24,25]. Patient education plays an important role for achieving an adequate bowel preparation and, therefore, for the success of colonoscopy. A questionnaire after regular instructions, a dietician-designed recipe, cell phone message reminding, education through multimedia, and personalized patient education were reported to enhance the effect of bowel preparation and decrease the rate of poor colon preparation. Li et al. [26] in their meta-analysis evaluated the effect of reinforced education (RE) by short message service (SMS) on the bowel preparation quality of patients undergoing colonoscopy. A total of seven RCTs with 5889 patients were subjected to meta-analysis. The rate of adequate bowel preparation in the SMS group (81.7%) was significantly higher than that in the control group (75.7%) (RR: 1.10, 95% CI: 1.03–1.17, *p* < 0.01). Four studies suggested that RE by SMS significantly reduced the non-attendance rate of patients for scheduled colonoscopy (RR: 0.74, 95% CI: 0.56–0.99, *p* < 0.05). Seven meta-analyses have been already conducted to compare the adequacy of bowel preparation in patients who received enhanced instructions and patients who received standard ones. All of them demonstrated that enhanced instructions are useful to improve the quality of BP and, at the same time, to increase ADR [27]. However, none of them focused specifically on visual booklets. In our meta-analysis, we considered six studies, which confirmed that booklets may enhance the bowel preparation, with an OR of 2.31. A recent meta-analysis found that newly designed booklets were beneficial to improving preparation, but the authors concluded that the evidence was poor due to the number of included studies, which was lower than that in our study [7]. At the end, in our study, we demonstrated that booklets are useful to enhance bowel preparation. Sub-analyses showed some interesting results. Outpatients seemed to benefit more from booklets. Since it has been demonstrated that education of the patient is pivotal to increase the quality of preparation [28], it is possible that the nurse care of inpatients in explaining the most correct way to assume preparation could have improved the quality and minimized the effect of the booklet [29]. Moreover, since senna shows slightly worse success than PEG [30], booklets may have optimized the results in these patients. Indeed, the positive effect of booklets was recorded when the preparation was not based on PEG.

On the other hand, the similar ADR in the two groups could be justified by the fact that few studies (only three) were included in such sub-analysis. Interestingly, the withdrawal time was shortened in the booklet than in the control group. A possible explanation might be that, since the preparation was better, the endoscopists had to spend less time cleaning the fecal residues during this phase.

Many reasons may explain the high level of heterogeneity among studies, including different settings and type of preparation, which have been partially solved by subgroup analysis. Sex could be a further source of heterogeneity, since one study [18] enrolled mainly male patients. Unfortunately, this could not be solved since studies did not provide results separately for males and females.

## 5. Conclusions

In conclusion, the present meta-analysis demonstrated that booklets are useful to improve bowel preparation. Outpatient settings and non-PEG-based formulations could benefit more from booklets. This evidence confirms the efficacy of booklets, but further confirmatory studies are needed to standardize their use as well as to achieve solid evidence for some outcomes, such as ADR.

## Figures and Tables

**Figure 1 jcm-12-04377-f001:**
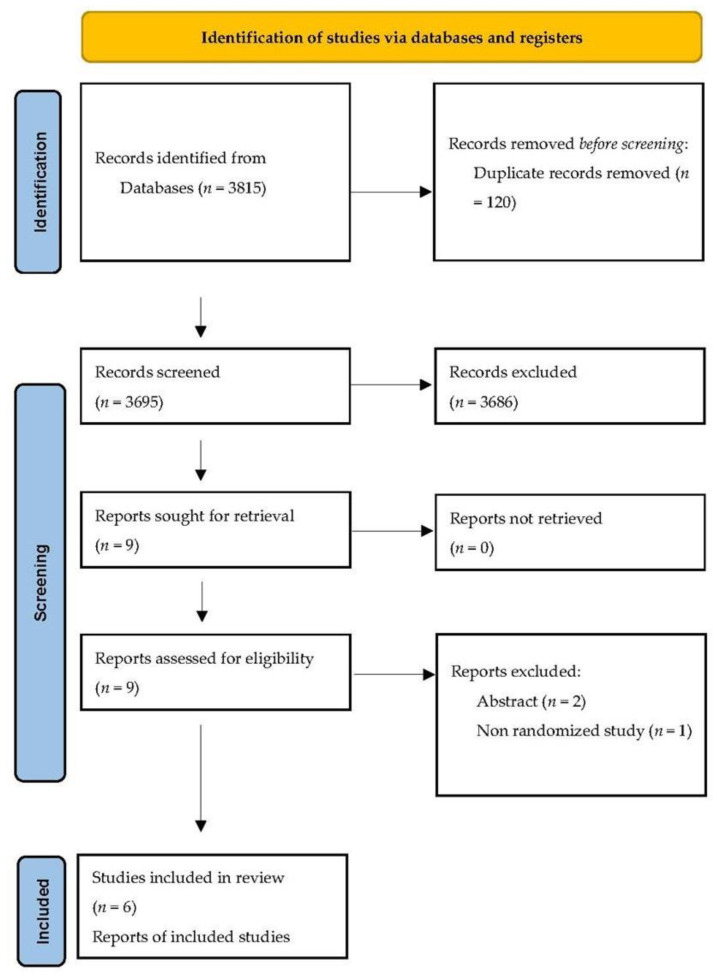
Process of study selection and exclusion.

**Figure 2 jcm-12-04377-f002:**
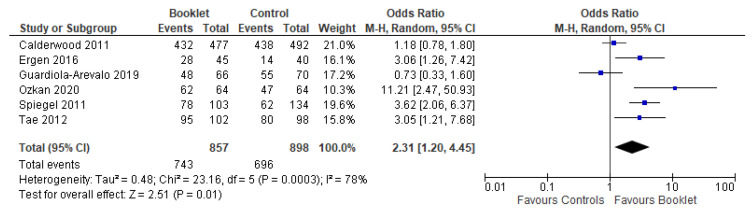
Forest plot reporting the effectiveness of booklets in achieving optimal bowel preparation.

**Figure 3 jcm-12-04377-f003:**
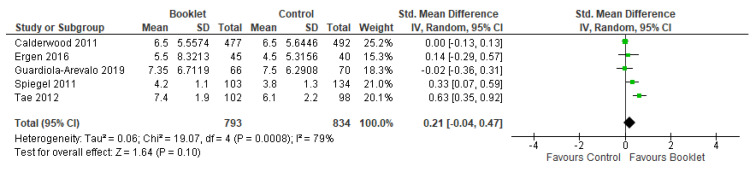
Weighted mean difference for score across studies.

**Figure 4 jcm-12-04377-f004:**
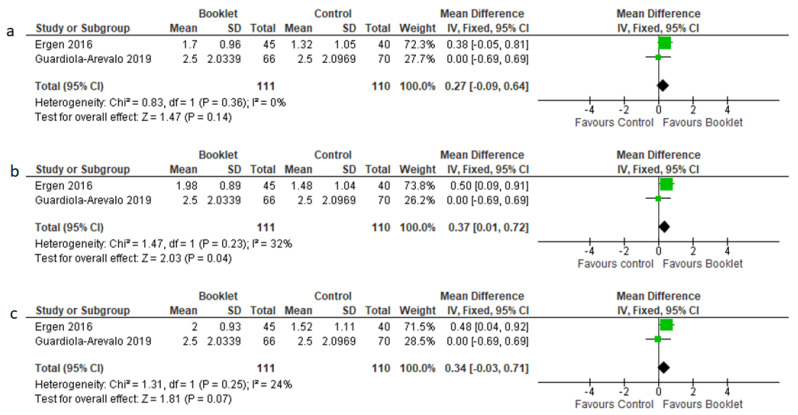
Standardized mean difference in colon cleansing scores, as assessed in left colon (**a**), transverse colon (**b**) or right colon (**c**).

**Figure 5 jcm-12-04377-f005:**
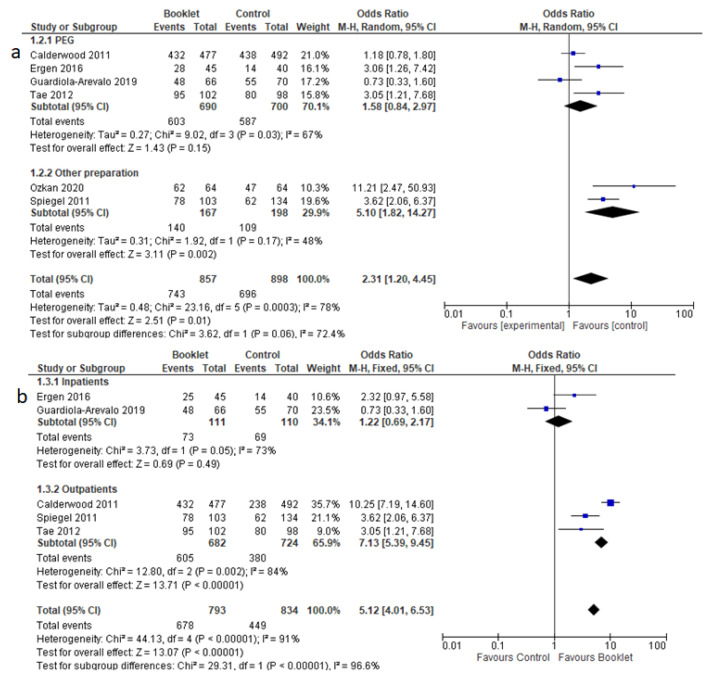
Sub-analysis of quality of bowel cleansing according to the type of preparation agent (**a**) and inpatient/outpatient setting (**b**).

**Figure 6 jcm-12-04377-f006:**
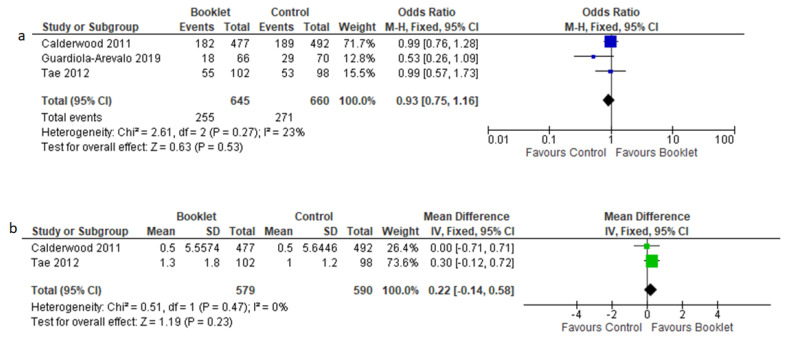
Adenoma detection rate (**a**) and mean number of polyps per patient (**b**) in booklet and control groups.

**Figure 7 jcm-12-04377-f007:**
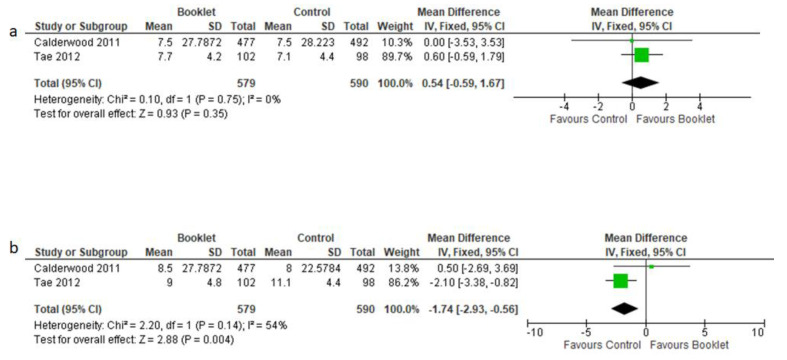
Standardized mean difference of insertion (**a**) and withdrawal time (**b**). Time is expressed as minutes.

**Table 1 jcm-12-04377-t001:** Main characteristics of included studies.

Study	Country, Year of Publication	Mean Age	Sex	Setting	Included Patients	Exclusion Criteria	Type of Preparation	Preparation Scale
Calderwood [14]	USA, 2011	57.1 ± 7.3 (standard of care) 57.3 ± 8.0 (booklet)	42% male	Outpatients	Screening colon cancer	Inflammatory bowel disease, colon infections, incomplete colonoscopy, preparation other than PEG	4L PEG ± bisacodyl	Boston
Ergen [15]	USA, 2016	58 ± 13 (standard of care) 57 ±15 (booklet)	male 64% in booklet group, 60% in control group	Inpatients	Low GI bleeding, anemia, diarrhea	Inflammatory bowel disease, dementia	4L PEG, split	Boston
Guardiola Arevalo [16]	Spain, 2019	66.2 (standard of care) 63.3 (booklet)	Male 51.43% (controls), 51.5% (booklet)	Inpatients	NA	Previous bowel resection, blindness, inflammatory bowel disease, dementia, previous colonoscopy 3 years before	4L PEG, 30% split	Boston
Ozkan [17]	Turkey, 2020	61.6 ± 12.2 (standard of care) 56.6 ± 10.9 (booklet)	Male 65.6% (booklet), 70.3% (controls)	Outpatients or in surgery ward	NA	NA	Sodium phosphate + enema	Derived from Johnson D, GIE 2014
Spiegel [18]	USA, 2011	60 ± 12.9 (standard of care) 60 ± 10.7 (booklet)	Male 98% (controls), 96% (booklet)	Outpatients	Colon cancer screening or surveillance, anemia, abdominal pain, hematochezia, inflammatory bowel disease, constipation, diarrhea	NA	Magnesium citrate, sodium phosphate, PEG	Likert 1–6
Tae [19]	South Korea 2012	47.6 ± 9.2 (standard of care) 48.6 ± 8.8 (booklet)	Male 68.9% (controls), 71.6% (booklet)	Outpatients	Colon cancer screening	Inflammatory bowel disease, colon infections, previous surgery, constipation, diarrhea	4L PEG split	Boston and UPAS

GI: gastrointestinal; NA: not available; PEG: polyethylene glycol.

**Table 2 jcm-12-04377-t002:** Summary of measures and GRADE evidence.

Outcome	No. of Participants (No. of Studies)	Relative Effect (95% CI)	Certainty of the Evidence (GRADE)	Conclusion
Adequate colon cleansing	1755 (6)	OR 2.31 (1.20–4.45)	●●●○ Moderate	Use of booklet improves the quality of bowel preparation
Total preparation score	1627 (5)	SMD 0.21 (−0.04–0.47)	●●●○ Moderate	Use of booklet does not improve the total score of bowel preparation
Preparation score per segment	221 (2)	Transverse colon: MD 0.37 (0.01–0.72) Right colon: MD 0.27 (−0.09–0.64) Left colon: MD 0.34 (−0.03–0.71)	●●○○ Low	Booklet improves quality of preparation in transverse colon only
Adequate preparation according to type of preparation	PEG: 1390 (4) Others: 365 (2)	PEG: OR 1.58 (0.94–2.97) Other: OR 5.10 (CI 1.82–14.27)	●●○○ Low	Use of booklet improves preparation when this is based on sodium phosphate but not on PEG
Adequate preparation according to setting	Inpatients: 221 (2) Outpatients: 1406 (3)	Inpatients: OR 1.22 (0.69–2.17) Outpatients: OR 7.13 (5.39–9.45)	●●○○ Low	Ambulatory patients benefit of booklet more than inpatients
Adenoma detection rate	1305 (3)	OR 0.93 (0.75–1.16)	●●●○ Moderate	Booklets do not improve adenoma detection rate
Mean number of polyps per patient	1169 (2)	MD 0.22 (−0.14–0.58)	●●○○ Low	Booklets do not increase the mean number of polyps detected per patient
Insertion time	1169 (2)	MD 0.54 (−0.59–1.67)	●●○○ Low	Insertion time is the same between controls and booklet group.
Withdrawal time	1169 (2)	MD −1.74 (−2.93 to −0.56)	●●○○ Low	Withdrawal time is longer when booklet is not given

The total number of black dots correlates with the Certainty.

## Data Availability

Not applicable.

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
