# Peer review of "Effect of Visual Booklets to Improve Bowel Preparation in Colonoscopy: Systematic Review with Meta-Analysis"

_jcm, 2023, doi:10.3390/jcm12134377_

Round 1

Reviewer 1 Report

This a metanalysis of the use of leaflets to improve colonoscopy preparation. It has been reasonably well written in English with sufficient references.  The methods including statistics are sound. However, the I2 score of 79% is alarmingly high. 

Additionally, the 6 studies selected seem to be skewed to mainly male patient and this should be mentioned. The Spiegel study is basically a male study. Males maybe better visual learners which should be discussed with a reference. 

Figure 2. Forest plot reporting the effectiveness of booklets in achieving optimal boerl preparation. (Spelling error) Also, what did you use as a score to consider optimal bowel prep? How did you convert the Likert scores to Boston prep scores?

Figure 4,6 and 7 Might not be needed and just stated in the text. since it's not significant.

Your discussion section starts very similar to the introduction and redundant language should be removed.

See above.

Author Response

This a metanalysis of the use of leaflets to improve colonoscopy preparation. It has been reasonably well written in English with sufficient references.  The methods including statistics are sound. However, the I2 score of 79% is alarmingly high.

Additionally, the 6 studies selected seem to be skewed to mainly male patient and this should be mentioned. The Spiegel study is basically a male study. Males maybe better visual learners which should be discussed with a reference.

We agree with the comment of the reviewer about I2 score. Regarding heterogeneity, an attempt was made to limit this drawback by carrying out sub-analyses (according to the setting and the type of preparation), as suggested by the Cochrane indications.

We thank the reviewer for pointing out the sex as a possible confounding factor. Despite we agree that this could be a further source of heterogeneity, unfortunately it cannot be solved since studies did not provide separate results between males and females. On the other hand, we cannot be sure that males could be better booklet readers than females. Indeed, most studies showed that male sex is an independent predictor of insufficient preparation (Laurie BD, Teoh MMK, Noches-Garcia A, Nyandoro MG. Colonic bowel prep and body mass index: does one size fit all? A multi-centre review. Int J Colorectal Dis. 2022;37(12):2451-2457).

Figure 2. Forest plot reporting the effectiveness of booklets in achieving optimal boerl preparation. (Spelling error) Also, what did you use as a score to consider optimal bowel prep? How did you convert the Likert scores to Boston prep scores?

We corrected the “boerl” typing mistake. Adequate preparation was considered for Boston if total points 6 with at least 2 point for each bowel segments and, for Likert, if ≥3 points. Since we used standardized mean difference for this meta-analytical analysis, this solved the heterogeneity between scores.

Figure 4,6 and 7 Might not be needed and just stated in the text. since it's not significant.

Indeed, despite being not significant, the message given by these figures could still be relevant and, furthermore, a graphical representation of the results may always be helpful for the readers to make more eye-catching the results of the present study.

Your discussion section starts very similar to the introduction and redundant language should be removed.

According also to the comment of reviewer 2, we changed the first sentences of the Discussion and we summarized the key concepts about the necessity of a good preparation for a high-quality colonoscopy.

Reviewer 2 Report

This study reported the effect of visual booklets to improve bowel preparation through a systematic review. The statistical analysis of the study was well performed and the manuscript was well written.

As we know, quite several clinical studies have been published in the last 10 years on using different methods, such as brochures, application programs, WeChat mini programs, etc, to improve the quality of bowel preparation, most methods being effective.

The results of this study reported that outpatient and non-PEG-based preparations seem to benefit more from booklets.

My detailed concerns are shown below.

1My major concern is the innovation of the research and whether it provides useful knowledge addition to this topic area. Furthermore, polyethylene glycol (PEG)-based bowel preparations are the most frequently used laxative regimens for colonoscopy worldwide.

2Some of the details of writing: for example, some sentences in the abstract should start with a capital letter.

3The first paragraph of the discussion requires a brief summary of the research results.

Well.

Author Response

This study reported the effect of visual booklets to improve bowel preparation through a systematic review. The statistical analysis of the study was well performed and the manuscript was well written.

As we know, quite several clinical studies have been published in the last 10 years on using different methods, such as brochures, application programs, WeChat mini programs, etc, to improve the quality of bowel preparation, most methods being effective.

The results of this study reported that outpatient and non-PEG-based preparations seem to benefit more from booklets.

My detailed concerns are shown below.

1、My major concern is the innovation of the research and whether it provides useful knowledge addition to this topic area. Furthermore, polyethylene glycol (PEG)-based bowel preparations are the most frequently used laxative regimens for colonoscopy worldwide.

We agree that PEG is the most used laxative agent in clinical practice and the most recommended one in guidelines. However, this does not mean that other preparations should be completely discarded, since they may can come in handy in some contexts, such as patients hard to prepare or subjects unwilling to drink large amounts of fluids. Therefore, we believe that the results of the present meta-analysis may give some information in this regard.

2、Some of the details of writing: for example, some sentences in the abstract should start with a capital letter.

All sentences in the Abstract started with a capital letter.

3、The first paragraph of the discussion requires a brief summary of the research results.

We added a paragraph in the discussion summarizing the research results.

Round 2

Reviewer 2 Report

This is my second review of this manuscript. I really appreciate the effort that the author and the research team have made to respond to my queries. With the revised manuscript I have no additional concerns or comments.

Some minor handwriting errors need to be corrected, such as punctuation after the word “databasesoutcomes” and so on?